# Draft, Verify, & Improve
# Toward Training-Aware Speculative Decoding

## Abstract

Autoregressive (AR) decoding is a major latency bottleneck for large language models. Speculative decoding (SD) accelerates AR by letting a drafter propose multi-token blocks that a verifier accepts or rejects. However, many SD systems require heavy offline training or extra components. These choices raise data/compute cost and can yield brittle drafters under distribution drift. We introduce *Draft, Verify, & Improve (DVI)*, a training-aware self-speculative framework that combines inference with continual online learning. We partition an LLM into a drafter and a verifier, and during generation, verifier accept/reject decisions are converted into supervision signals and used to update the drafter head. A simple $KL{\rightarrow}RL$ schedule bootstraps calibration via online distillation and then adds reward-masked cross-entropy with a on-policy policy-gradient term, preserving lossless, single model deployment. On Spec-Bench, DVI achieves a $2.16\times$ wall-time speedup, on par with SoTA approaches like EAGLE-2, while orders of magnitude less data for training, and ablations show that DVI outperforms KL-only online distillation. DVI demonstrates that *training-aware* self-speculation can deliver state-of-the-art, lossless speedups with minimal training overhead.

Code will be released upon publication.

## 1 Introduction

Large Language Models (LLMs) are continuing to rise in popularity, finding new use-cases ranging from powering conversational assistants to web agents. However, models continue to suffer high latency due to their autoregressive (AR) nature. AR decoding acts as the primary bottleneck for LLM inference as each token depends on the previous one, causing end-to-end latency to scale sequence length and per-step compute.

Speculative decoding (SD) addresses this by proposing multiple tokens at once instead of one token at a time (Chen et al., 2023; Leviathan et al., 2023). SD uses a lightweight drafting model, and uses the larger, high-fidelity target model to verify or reject each drafted token. The accepted tokens are then committed and generated, preserving target-model correctness while reducing wall-clock time. While SD has exploded in popularity, many SD works still face practical limitations. Speedups are dependent on high token acceptance rates, which in turn hinge on the drafter's fidelity. If the drafted tokens diverge, the verifier takes over, and falls back to AR decoding, plus the added drafting computation.

In traditional SD, many works train a dedicated drafting model, or distill a drafting model from the larger target model (Chen et al., 2023; Leviathan et al., 2023). Other works explore self-speculation, where the model itself is partitioned into drafting layers and verification layers (Zhang et al., 2023; Liu et al., 2024a). Regardless of which approach, most techniques rely heavily on a robust training phase to train the drafting model. This heavy offline drafter training introduces practical gaps for SD.

Many SD pipelines require training a drafting model for multiple epochs over large datasets, or intensive model distillation from full teacher distributions over data (Ankner et al., 2024; Cai et al., 2024; Li et al., 2024a;b; Zhang et al., 2023). Even training a lightweight adapter requires hours of training (Liu et al., 2024a).

This reliance on heavy offline training enables drafter brittleness under distribution shift. As conversation, task, or traffic drifts, the fixed drafters lose acceptance, killing any potential speedup, if the distribution is not covered by the training data (Liu et al., 2024b). These slowdowns compound in traditional SD settings, where the pipeline relies on separate drafter and verifier models, due to the added computational cost of running both models, and the ensuring KV cache overhead, system complexity, and increased memory demands.

Practical speed is therefore conditional on the cost of *reaching and maintaining* a well-calibrated drafter. Retraining large tree-based or multi-head drafters on millions of prompt exposures can quickly dominate the engineering and compute budget, especially when models are updated frequently or specialized to new applications.We therefore treat training/maintenance overhead as a first-class constraint rather than a secondary metric. We seek not only wall-clock speedups, but speedups that can be obtained cheaply and continuously as traffic drifts, without requiring separate offline retraining pipelines.

Therefore, we propose Draft, Verify, & Improve (DVI), a training-aware self-speculative decoding framework. DVI splits a single backbone model at an intermediate layer, attaching a drafter head and a frozen verifier head. This partitioning creates a computationally efficient drafting model and verifying model. At inference, the drafter proposes multi-token blocks which the verifier accepts or rejects.

Whereas most SD pipelines will just commit the accepted tokens and move on, DVI instead treats these commit decisions as learning signals: the drafter is updated online from accept/reject feedback while the verifier remains unchanged. This preserves a one-model serving geometry with no auxiliary drafter or extra KV cache while enabling continual drafter adaptation to live traffic.

During generation, the drafter proposes $k_{\text{spec}}$ tokens from prefix tokens $h_k$; the frozen verifier evaluates the same prefix from $h_L$ and commits the longest agreeing prefix between the drafter and verifier. The frozen verifier ensures speculation remains lossless. DVI logs the per-token state at $h_k$, denoting a accept or reject, into a lightweight buffer and performs small, frequent updates to the drafter using a *KL→RL schedule*: a KL-guided warmup to the verifier distribution followed by reward-masked cross-entropy and an on-policy policy-gradient term (Williams, 1992). Training can be done in an online manner, as it mirrors the inference workflow of drafting and verifying, minimizing train/serve skew.

We evaluate DVI on Spec-Bench, a public SD benchmark suite spanning translation, summarization, QA, math, and retrieval-augmented generation. DVI achieves near-$2\times$ average wall-time speedups with a single pass over a small prompt stream. Acceptance improves steadily during online updates, and the method maintains strong performance without any additional offline training or auxiliary models.

We perform ablations to verify that our training pipeline is robust. We demonstrate that optimizing the drafter purely by distillation or purely by sparse rewards is insufficient. A KL/distillation-only variant, performing online KD from the frozen verifier, creates a speedup, but falls short of matching DVI performance.

These results motivate DVI's *KL→RL* schedule: an online KD warmup that calibrates the shallow drafter in the verifier's logit space, followed by a on-policy correction that assigns credit only where speculation succeeds. This combination overcomes both the KL plateau and the instability of sparse-reward training, directly optimizing acceptance where it matters.

**Contributions:**

- We propose DVI, a self-speculation method with a frozen verifier and an online-learned drafter head that converts commit decisions into self-supervision.

- DVI offers a data-efficient, cheap method to train a SD model. We create competitive speedups using a small number of live prompts, with no separate offline dataset or long pretraining.

- We validate DVI's effectiveness by performing experiments with Spec-Bench, demonstrating competitive speedups compared to other SoTA SD methods.

The paper is structured as follows: Section 2 introduces related works. Section 3 presents the DVI pipeline, objective, and training schedule. Section 4 reports speedups, comparisons against other SD methods, and ablations. Section 5 concludes the paper.

## 2 RELATED WORK

### 2.1 SPECULATIVE DECODING BASICS

Traditional modern speculative decoding (SD) accelerates autoregressive generation by letting a small draft model propose a block of $k$ tokens and a large target model verify them in parallel. The target model then commits the longest agreeing prefix and continues generation. In this setting, the committed tokens from the verifier are exactly from the target model's decoding distribution (e.g., greedy or sampling) — creating "lossless" speculation (Leviathan et al., 2023; Chen et al., 2023).

Liu et al. (2024b), periodically fine-tunes an external drafter model online via knowledge distillation (KD) (Liu et al., 2024b). By aligning to live query distributions, it shows that continual adaptation is possible, albeit with an auxiliary drafter and a pretraining warm-start.

### 2.2 SELF-SPECULATION

Instead of a separate drafting model, self-speculation splits a single backbone model into shallow drafting layers, and deep verification layers. Zhang et al. (2023) introduced self-speculation for LLMs by skipping or shortening intermediate layers to form a fast internal drafter and then running the full model to verify.

This simpler approach removes the complexity of managing multiple models, finding suitable drafting models for target models, and reduces computation.

Later work improves upon this, with more complex and flexible architectures. For example, Liu et al. (2024a) achieves self speculation by adding a lightweight adapter over the shallow subnetwork and introducing dynamic early-exit during drafting to curb drafter latency when confidence is low.

### 2.3 OTHER ACCELERATION FAMILIES

Instead of speculating, Cai et al. (2024) augments the backbone model with multiple time-independent heads that predict multi-step continuations. The resulting branches are verified with tree-attention verifier every step. Ankner et al. (2024) replaces Medusa's independent heads with sequentially dependent heads, improving draft accuracy (and thus acceptance) under the same verification framework.

Li et al. (2024a) drafts features (second-to-top-layer states) one step ahead and uses the target LM head to form tokens, producing well-calibrated draft trees. Li et al. (2024b) adds context-aware dynamic trees, yielding improved speedups. While these approaches create massive speedups, they also require significant amounts of training, even relative to approaches like Medusa.

## 3 DVI: DRAFT → VERIFY → IMPROVE

This section formalizes *Draft, Verify, & Improve (DVI)* for self-speculative decoding using a single backbone model with a LoRA-parameterized draft head.

### 3.1 PRELIMINARIES AND NOTATION

Consider a decoder-only language model with transformer layers indexed $0{:}L$. We choose a split index $k$ with $0 < k < L$ and write the shallow and deep hidden states as in equation 1:

$$h_{k,t} = f_{0 \to k}(x_{0:t}), \qquad h_{L,t} = f_{k \to L}(h_{k,t}), \tag{1}$$

where $x_{0:t}$ is the token prefix, $f_{0 \to k}$ is the *draft path* (layers $0{\to}k$), and $f_{k \to L}$ is the *target path* (layers $k{\to}L$).

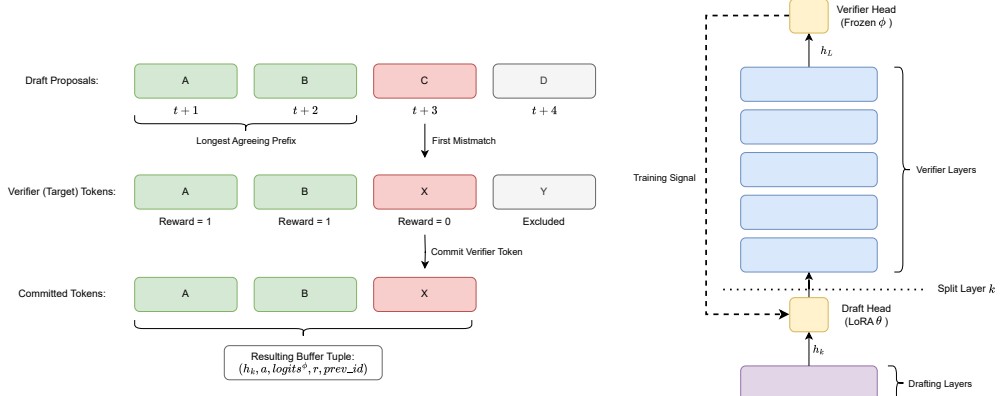

Figure 1: **Left:** Multi-token speculation, where the drafter proposes a block of tokens and the verifier accepts the longest agreeing prefix before emitting the first mismatch. We log one tuple per drafted position up to and including the first reject, $(h_k, a, \text{logits}^\phi, r, \text{prev\_id})$, with $r=1$ for accepted tokens and $r=0$ for the first reject. This converts verifier feedback into continual self-supervision. **Right:** DVI architecture, where the backbone is split at layer $k$, with shallow drafting layers (purple) feeding the LoRA draft head $p_\theta(\cdot \mid h_k)$ and deep verification layers (blue) feeding the frozen verifier head $p_\phi(\cdot \mid h_L)$. The logged tuples from the rollout buffer drive updates to the draft head, while the verifier and backbone remain fixed. This closes the loop between online speculation and training, ensuring adaptation without additional models or offline data.

We attach two vocabulary classifiers as in Eqs. equation 2 and equation 3:

$$p_\phi(\cdot \mid h_{L,t}) = \text{softmax}(W^{(V)} h_{L,t}) \quad \text{(target path; \textit{frozen})}, \tag{2}$$

$$p_\theta(\cdot \mid h_{k,t}) = \text{softmax}\Big((W^{(S)} + \gamma_s A_s B_s) h_{k,t}\Big) \quad \text{(draft path; \textit{trainable LoRA})}. \tag{3}$$

Here $W^{(S)}$ is a frozen base projection at the draft path output, and $(A_s, B_s)$ are trainable LoRA modules (Hu et al., 2021). Throughout training, *only* $\theta \equiv \{A_s, B_s\}$ is updated online; all backbone weights remain fixed. LoRA is applied only to the draft head at $h_{k,t}$.

We adopt the canonical longest-prefix verification used in SD: at each position $t$, the target path deterministically emits the next token under a fixed sampler. Like many other SD works, we employ *greedy* decoding as in equation 4:

$$y^\star_{t+1} = \arg\max_y p_\phi(y \mid h_{L,t}), \tag{4}$$

and the procedure is lossless because verification preserves the target sampler's output. Additionally, we consider only *single-sequence* verification (no tree search).

## 3.2 SELF-SPECULATIVE FACTORIZATION

As mentioned before, DVI is self-speculative: the backbone model is partitioned into a draft path (layers $0{\rightarrow}k$) and a target path (layers $k{\rightarrow}L$). The LoRA-augmented draft head at $h_{k,t}$ proposes tokens quickly; the frozen target head at $h_{L,t}$ verifies the tokens. This setup is similar to that of Kangaroo (Liu et al., 2024a), though without their further optimizations like dynamic drafting. This single-model setup avoids separate drafting and target models, the ensuing extra KV cache, and larger system level complexity, while providing a LoRA adapters for online learning.

Drafting a $k_{\text{spec}}$-token block requires one shallow forward; if $m$ tokens are accepted, the deep computation is amortized over those $m$ outputs in a single verification pass.

## 3.3 SPECULATIVE ROLLOUT AND LEARNING SIGNAL

We can see what one speculation and verification looks like given the notation above.

At decoding step $t$, we compute the shallow state once given the prefix tokens (commonly user input or training data), as in equation 5:

$$h_{k,t} = f_{0 \to k}(x_{0:t}). \tag{5}$$

The draft path rolls out up to $k_{\text{spec}}$ candidates as in equation 6:

$$\tilde{t}_{t+1:t+k_{\text{spec}}} \sim p_\theta(\cdot \mid h_{k,\cdot}), \tag{6}$$

Given the drafted candidates, we verify for each prefix length $i = 1, \ldots, k_{\text{spec}}$, as in equation 7:

$$h_{L,t+i-1} = f_{k \to L}(h_{k,t+i-1}), \qquad y^\star_{t+i} = \arg\max p_\phi(\cdot \mid h_{L,t+i-1}). \tag{7}$$

Let $m$ be defined as in equation 8:

$$m = \max \left\{ i \in \{0, \ldots, k_{\text{spec}}\} : \tilde{t}_{t+j} = y^\star_{t+j} \text{ for all } j \leq i \right\}. \tag{8}$$

We then commit the $m$ agreeing tokens and reject everything at and after the first mismatch by emitting $y^\star_{t+m+1}$. Decoding then continues autoregressively from the new prefix.

The resulting accept/reject outcomes yield a clean, low-variance signal. For drafted positions up to and including the first reject, we log to an *online replay buffer* as in equation 9:

$$\left( h_{k,t+i-1},\ a_{t+i} = \tilde{t}_{t+i},\ \text{logits}^\phi_{t+i},\ r_{t+i},\ i \right), \tag{9}$$

where $\text{logits}^\phi_{t+i}$ are target-path logits, and $r_{t+i}$ is defined in equation 10 as

$$r_{t+i} = \begin{cases} 1, & 1 \leq i \leq m \quad \text{(accepted)}, \\ 0, & i = m+1 \quad \text{(first reject)}, \\ \text{undefined}, & i > m+1 \quad \text{(counterfactual; not verified)}. \end{cases} \tag{10}$$

We exclude $i > m+1$ from supervised terms to avoid counterfactual bias. The buffer mirrors inference (same $k_{\text{spec}}$ and commit rule), reducing train/serve skew, and generalizing to online learning.

## 3.4 OBJECTIVES AND UPDATE SCHEDULE

To prevent an RL-cold start with low acceptance, early updates imitate the target path (online KD) to stabilize gradients in the low-rank subspace; later, the draft path optimizes acceptance on observed traffic. We implement a composite objective with a KL-to-RL schedule acting on the LoRA adapters $\theta$.

For a minibatch $\mathcal{B}$ sampled from the online buffer, we minimize the composite loss in equation 11:

$$\mathcal{L}_{\text{fast}} = \lambda_{\text{pg}}\,\mathcal{L}_{\text{pg}} + \lambda_{\text{kl}}\,\text{KL}\big(p_\theta \,\|\, p_\phi^{(\tau)}\big) + w_{\text{ce}}\,\mathcal{L}_{\text{CE}} - w_{\text{ent}}\,\mathcal{H}[p_\theta], \tag{11}$$

where $p_\theta(\cdot \mid h_{k,t}) = \text{softmax}\big((W^{(S)} + \gamma_s A_s B_s)h_{k,t}\big)$ and $p_\phi^{(\tau)} = \text{softmax}(\text{logits}^\phi/\tau)$.

The reward-masked term is defined in equation 12:

$$\mathcal{L}_{\text{pg}} = -\frac{1}{|\mathcal{P}|} \sum_{i \in \mathcal{P}} \log p_\theta(a_{t+i} \mid h_{k,t+i-1}) \tag{12}$$

This term uses only *accepted* positions $\mathcal{P}$ to focus credit where speculation succeeded, and $\text{KL}(p_\theta \| p_\phi^{(\tau)})$ performs online KD for calibration.

Once the cold start is avoided, we add a light, *on-policy* correction on fresh tuples, as in equation 13:

$$\mathcal{L}_{\text{policy}} = w_{\text{rl}}\,\mathbb{E}_{(s,a,r) \sim \text{on-policy}}\big[-(r-b)\log p_\theta(a \mid s)\big] + \beta(t)\,\text{KL}\big(p_\theta \,\|\, p_\phi\big), \tag{13}$$

where $b$ is a baseline for variance reduction (we use an EMA of recent rewards) and $\beta(t)$ gently decays to retain calibration. We include both accepted and first-reject positions; positions beyond the first reject are excluded (counterfactual).

We anneal the relative weights over wall-clock updates $t$ according to the schedule in equation 14:

$$(\lambda_{\text{pg}}, \lambda_{\text{kl}})(t) = \begin{cases} (0,\ \lambda_0), & t < T_{\text{warmup}}, \\ \left(\dfrac{t - T_{\text{warmup}}}{T_{\text{ramp}}}\lambda_{\text{pg}}^{\max},\ \lambda_0 - \dfrac{t - T_{\text{warmup}}}{T_{\text{ramp}}}(\lambda_0 - \lambda_{\text{kl}}^{\min})\right), & \text{ramp}, \\ (\lambda_{\text{pg}}^{\max},\ \lambda_{\text{kl}}^{\min}), & \text{after}. \end{cases} \tag{14}$$

Warmup emphasizes online KD to avoid unstable gradients in a misaligned subspace; the ramp increases reward-masked learning that directly raises acceptance on the observed traffic.

## 4 EXPERIMENTS

### 4.1 EXPERIMENTAL SETUP

All experiments use the Spec-Bench benchmark (Xia et al., 2024). Spec-Bench is a public speculative decoding benchmark that measures model speedups and accepted tokens over six evaluation settings (MT-Bench, Translation, Summarization, QA, Math, and RAG). Since our training data differs from Spec-Bench in both task distribution and data source, Spec-Bench acts as an out-of-training-distribution (OOD) evaluation metric for all the tested speculative decoding strategies.

We perform a series of experiments using different sizes of the Vicuna (Zheng et al., 2023) model series, as requested by the Spec-Bench benchmark. We train and benchmark the Vicuna-7B, Vicuna-13B and Vicuna-33B models with the serving configuration fixed across all model sizes and methods (identical tokenizers, decoding policies, context limits, and so on).

We note that while the Vicuna series of models is now outdated, the Spec-Bench benchmark explicitly defines Vicuna as the base model to test with. Competing methods like Medusa (Cai et al., 2024), Hydra (Ankner et al., 2024), EAGLE-1 (Li et al., 2024a), and EAGLE-2 Li et al. (2024b) provide implementations that are based on Vicuna models. We mirror their model selection choice order to guarantee a fair comparison between DVI and the competing methods.

Table 1: Comparison of training data budgets across speculative decoding methods. DVI uses orders of magnitude fewer prompt exposures than prior approaches.

| Method | ShareGPT Samples | Epochs | Prompt exposures | Optimiser steps | Relative budget |
|---|---|---|---|---|---|
| DVI (our work) | 2,000 | 1 | 2,000 | 2,000 | $1\times$ |
| Medusa (Cai et al., 2024) | 60,000 | 2 | 120,000 | $\approx 945$ | $\sim 60\times$ more |
| Kangaroo (Liu et al., 2024a) | 60,000 | 20 | 1,200,000 | $\approx 4,700$ | $\sim 600\times$ more |
| EAGLE (Li et al., 2024a) | 60,000 | 40 | 2,400,000 | $\approx 300,000$ | $\sim 1,200\times$ more |

For the DVI model, we vary the split layer $k$ depending on the model size. For the 7B model, we split the model at layer 2 such that layers 3 through 32 are the verifier. For Vicuna-13B, we split at layer 3, and for Vicuna 33B, we split at layer 5. These choices were made to mirror other layer selection strategies found in other self-speculation works like Kangaroo (Liu et al., 2024a). Besides the split layer, all other hyperparameters are held the same across the various model sizes.

For all DVI models, the verifier is the frozen baseline weights, guaranteeing lossless speculation. In the experimental setting, we use a drafting proposal depth of 4 ($k_{\text{spec}}=4$). We train the DVI model for 2000 steps over 2000 prompts, such that the model sees each prompt only once.

Competing methods are taken *as-is* from their public checkpoints and implementations surfaced through Spec-Bench and Hugging Face. We do not retrain baselines; where the harness exposes a method knob (e.g., draft depth), we use the recommended defaults.

We train DVI on 2,000 samples of ShareGPT, aligning our training data with other methods in the literature. Methods like EAGLE, Medusa, Hydra, and Kangaroo all train their models on ShareGPT as well. We illustrate in Table 1 that the competing methods are trained for orders of magnitude longer than DVI, in an offline setting. Comparatively, DVI sees a fraction of the prompts that the other methods do, making inherently cheaper to train and more data efficient than other methods.

It is important to note not all baselines were available for all model sizes. EAGLE-3 did not provide a Vicuna-7B or Vicuna-33B implementation, but it does provide a Vicuna 13B implementation (Li et al., 2025).

Following Spec-Bench, we report mean accepted tokens (MAT) per verification step, and wall-time speedup versus the baseline model's standard autoregressive decoding. Note that MAT is not a one-to-one predictor of model speedup. Having a deeper drafting model for example, will yield a higher MAT, but at the cost of added computation, risking a lower speedup. Methods that propose more tokens will inherently have higher MAT, but it may also induce extra computation that offsets this advantage.

Table 2: Spec-Bench comparison across speculative decoding methods for 7B, 13B, and 33B models. Each cell shows mean accepted tokens (MAT) and walltime speedup; the rightmost column reports the overall average speedup.

| Size | Method | MT Bench | | Translation | | Summarization | | QA | | Math | | RAG | | Avg. |
|------|--------|-----|--------|-----|--------|-----|--------|-----|--------|-----|--------|-----|--------|------|
| | | MAT | Speedup | MAT | Speedup | MAT | Speedup | MAT | Speedup | MAT | Speedup | MAT | Speedup | |
| 7B | EAGLE-2 | 4.75 | **2.64**× | 3.22 | 1.73× | 3.96 | **2.15**× | 3.70 | 1.96× | 4.73 | **2.59**× | 4.09 | 2.02× | **2.18**× |
| 7B | EAGLE-1 | 3.83 | 2.38× | 2.84 | 1.72× | 3.32 | 2.03× | 3.12 | 1.88× | 3.87 | 2.39× | 3.27 | 1.88× | 2.05× |
| 7B | Hydra | 3.59 | 2.31× | 2.80 | 1.80× | 2.70 | 1.73× | 2.85 | 1.84× | 3.61 | 2.31× | 2.90 | 1.74× | 1.96× |
| 7B | Medusa | 2.51 | 1.91× | 2.13 | 1.59× | 2.02 | 1.53× | 2.09 | 1.59× | 2.50 | 1.88× | 2.10 | 1.48× | 1.66× |
| 7B | PLD | 1.69 | 1.61× | 1.10 | 1.03× | 2.72 | 2.54× | 1.37 | 1.14× | 1.86 | 1.59× | 1.72 | 1.85× | 1.62× |
| 7B | SpS | 2.33 | 1.62× | 1.46 | 1.10× | 2.44 | 1.65× | 2.17 | 1.45× | 2.20 | 1.46× | 2.31 | 1.63× | 1.48× |
| 7B | DVI (ours) | 3.07 | 1.97× | 3.53 | **2.24**× | 3.55 | 2.02× | 3.61 | **2.14**× | 3.04 | 2.02× | 3.53 | **2.58**× | 2.16× |
| 13B | EAGLE-3 | 5.98 | **1.85**× | 4.32 | 1.26× | 5.76 | **1.75**× | 4.80 | 1.54× | 5.82 | **1.80**× | 5.72 | 1.54× | **1.62**× |
| 13B | EAGLE-2 | 4.79 | 1.81× | 3.30 | 1.21× | 4.12 | 1.60× | 3.50 | 1.32× | 4.81 | 1.78× | 4.26 | 1.50× | 1.54× |
| 13B | EAGLE-1 | 3.89 | 1.76× | 2.94 | 1.29× | 3.44 | 1.60× | 2.91 | 1.33× | 3.92 | 1.74× | 3.53 | 1.49× | 1.53× |
| 13B | PLD | 1.62 | 1.03× | 1.11 | 0.82× | 2.51 | 1.45× | 1.17 | 0.80× | 1.84 | 1.08× | 1.81 | 1.16× | 1.04× |
| 13B | DVI (ours) | 3.08 | 1.48× | 3.59 | **1.49**× | 3.51 | 1.59× | 3.43 | **1.59**× | 3.09 | 1.56× | 3.88 | **1.67**× | 1.56× |
| 33B | EAGLE-2 | 4.26 | **2.93**× | 3.17 | 2.08× | 3.82 | 2.48× | 3.25 | 2.25× | 4.78 | **3.30**× | 3.63 | 2.31× | **2.56**× |
| 33B | EAGLE-1 | 3.54 | 2.70× | 2.80 | 2.00× | 3.25 | 2.37× | 2.83 | 2.11× | 3.89 | 2.93× | 3.07 | 2.20× | 2.39× |
| 33B | Hydra | 3.48 | 2.58× | 2.77 | 2.03× | 2.77 | 1.98× | 2.79 | 2.14× | 3.68 | 2.76× | 2.97 | 2.03× | 2.26× |
| 33B | Medusa | 2.44 | 2.00× | 2.20 | 1.77× | 2.06 | 1.60× | 2.05 | 1.70× | 2.61 | 2.13× | 2.13 | 1.61× | 1.81× |
| 33B | PLD | 1.51 | 1.41× | 1.10 | 1.03× | 2.21 | 1.86× | 1.21 | 1.04× | 1.73 | 1.51× | 1.43 | 1.38× | 1.36× |
| 33B | DVI (ours) | 2.75 | 2.30× | 3.57 | **2.49**× | 3.71 | **2.54**× | 3.35 | **2.33**× | 3.37 | 2.67× | 3.23 | **2.73**× | 2.51× |

More implementation details are noted in Appendix A.

## 4.2 RESULTS

DVI achieves competitive walltime speedups with methods like EAGLE-2, while remaining lossless, and using orders-of-magnitude less training data accross all model sizes. It is important to highlight that while EAGLE-3 may have the greatest speedup on average, it is not strictly faster than DVI. As shown in Table 2, DVI consistently attains the highest speedup in Translation, QA, and RAG, outperforming EAGLE-2 and EAGLE-3 despite EAGLE having a $1000\times$ larger training budget as shown in Table 1. However, EAGLE-2 and EAGLE-3 do lead in MT Bench, Summarization, and Math.

At a more granular level, these results indicate that DVI is particularly effective for workloads with strong local lexical structure and retrieval grounding (assistant-style tasks), whereas deep tree drafting is more beneficial for long-range reasoning and structured planning.

We should not discount the role that training data plays. ShareGPT covers a wide range and variety of tasks, and the 2,000 samples we take from it are not necessarily evenly distributed. Therefore, DVI may perform better on tasks like MT Bench if exposed to more multi-turn conversations and so on.

Although EAGLE attains a higher mean accepted tokens (MAT), its average wall-time speedup is only marginally higher than DVI. This gap implies that DVI converts agreement into throughput more efficiently. With a shallow drafter and small proposal depth, DVI avoids the extra tree-building and verification work required by multi-head or tree-based approaches, yielding a higher speedup-per-accepted-token and explaining why similar end-to-end gains emerge despite lower MAT.

## 4.3 ABLATIONS

We isolate the contribution of each training signal in DVI by ablating the objective into three single-term variants:

1. KL-only, equivalent to online distillation.

2. PG-only, which is simply on-policy REINFORCE.

3. CE-only, reward-masked cross entropy.

For these ablations, all runs are on the Vicuna-7B model, with split layer $k = 2$ and proposal depth $k_{\text{spec}}=4$. The optimizer, batch size, data stream, and hardware are the same as our main experimental setup.

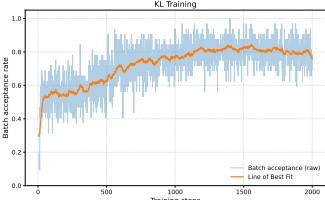
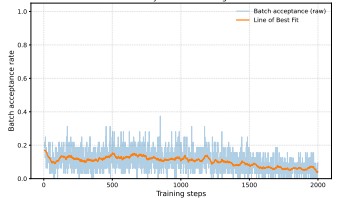
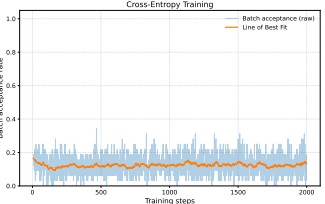

(a) KL-ONLY: stable, monotone gains.

(b) PG-ONLY: flat, noisy learning curve.

(c) CE-ONLY: flat learning curve.

Figure 2: **Objective ablations:** Batch acceptance rate vs. training steps. Curves computed on the same data stream, split, and $k_{\text{spec}}$ as the main setup.

We first observe the learning dynamics of the model, captured by the batch acceptance rate throughout training. We then benchmark the resulting models on Spec-Bench to get their end-state MAT and wall-time speedup. We define the batch acceptance rate to be the fraction of drafted tokens accepted in each optimization step (higher is better). Note that batch acceptance rate is not a perfect indicator model performance on Spec-Bench, as the speculative batches are directly from training on ShareGPT. Thus, models can attain higher batch acceptance without becoming robust on other distributions, like tasks in Spec-Bench.

Table 3: Objective ablations on Spec-Bench. Higher is better.

| Objective | Mean accepted tokens (MAT) | Speedup |
|-----------|----------------------------|---------|
| KL-ONLY | 1.933 | 1.435× |
| PG-ONLY | 0.035 | 0.341× |
| CE-ONLY | 0.039 | 0.335× |

**KL-Only:** We observe that online KD is sufficient to raise acceptance. The batch-acceptance curve increases steadily and smoothly across training (Fig. 2a), indicating consistent improvements in draft/verifier agreement. However, the the acceptance curve begins to level out around a 80% batch acceptance rate, indicating that there are limits to KL alone.

This is further verified by the end results, as the KL-only training attains the highest speedup among single-term objectives (Table 3), but still falls short of our full DVI pipeline. This demonstrates that because gradients are dense and low variance, KL alone can bootstrap a useful drafter, which empirically validates our KL-heavy warmup to avoid a cold start in RL.

**PG-Only:** We observe that sparse rewards and censored feedback hinder learning. Acceptance remains near zero and exhibits noise rather than an upward trend (Fig. 2b). The end-state MAT and speedup indicate the model is $\sim 3\times$ slower than a baseline model (Table 3).

With a frozen verifier and shallow draft, rewards are extremely sparse as only agreement with the verifier provides $r=1$. This "bandit with censoring" produces high-variance gradients and weak credit assignment. Without KD to keep logits calibrated, exploration drifts, further reducing agreement and compounding variance. These results support keeping PG as a light, on-policy correction after KD stabilization.

**CE-only:** We observe that reward-masked supervision is too weak without calibration. Similar to the PG-only objective, acceptance stays flat over training (Fig. 2c), and end-state MAT and speedup indicate a model $\sim 3\times$ slower than the baseline model (Table 3). Because only accepted tokens are labeled, the CE target distribution is censored and narrow. Similar to PG-only, the distribution-level guidance of KL is necessary to improve acceptance.

**Insights**  The ablations confirm that

1. KL-only reliably increases acceptance (dense, low-variance signal).
2. PG-only and CE-only struggle under sparse/censored feedback.

It should be note that many sampling-based SD methods and Online SD methods rely almost exclusively on KL / distillation for teaching the drafter (Liu et al., 2024b; Zhou et al., 2023). We demonstrate that in a greedy decoding setting, strict distillation is significantly slower than DVI's approach.

These results empirically justify DVI's staged objective as described in Section 3. The schedule avoids RL cold start, preserves losslessness under the same sampler, and keeps training cheap and robust.

## 5  CONCLUSION

We presented *Draft, Verify, & Improve (DVI)*, a training-aware self-speculation method that closes the loop between inference and learning. DVI splits a single backbone into shallow drafting and deep verifying paths, keeps the verifier frozen, and updates a lightweight LoRA drafter online from accept/reject feedback.

When evaluated on Spec-Bench, DVI delivers $2.16\times$ end-to-end speedups on average, matching methods like EAGLE-2, and surpassing SoTA methods like EAGLE-3 on Translation, QA, and RAG. DVI attains these gains with a dramatically smaller training budget: 2,000 online prompts versus the *millions* of prompt exposures used by other methods.

Ablation studies validate our training pipeline. Online KL (distillation) alone produces steady acceptance gains, but plateaus; PG-only and reward-masked CE-only objectives struggle due to sparse/censored rewards and poor calibration. Our proposed KL→RL schedule stabilizes early learning and adds targeted on-policy improvements, yielding the full DVI performance.

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

## A  EXPERIMENTAL SETUP AND REPRODUCIBILITY

**Hardware.**   All experiments were run on a single NVIDIA H100 GPU. To check hardware sensitivity, we repeated a subset of runs on an NVIDIA A40; qualitative conclusions and relative rankings were unchanged. Due to compute and time constraints, we did not replicate all experiments on different GPUs.

**Model and decoding configuration.**   All results use the Vicuna-7B backbone specified by SPEC-BENCH. We adopt the self-speculative split from the main text: the draft path comprises the shallow layers up to index $k=2$ and the target path comprises the remaining layers ($3\rightarrow L$). The drafter head is LoRA-parameterized and trainable; the verifier head and backbone are frozen. Unless otherwise stated, we use greedy decoding (temperature 0) for verification, proposal depth $k_{\mathrm{spec}}=4$, and the tokenizer and context limits mandated by the benchmark.

**Training configuration.**   For the DVI experiments, we train using 2,000 prompts from ShareGPT as training samples. The composite objective in 11 uses a CE weight $w_{\mathrm{ce}} = 0.07$, no entropy bonus ($w_{\mathrm{ent}} = 0$), and a KL term with initial weight $\lambda_{\mathrm{kl}} = \lambda_0 = 0.55$ annealed down to $\lambda_{\mathrm{kl}}^{\mathrm{min}} = 0.08$, with a KL-warmup window of 700 steps and warmup scale 1.2. The policy-gradient component in 13 is enabled with a maximum PG weight $\lambda_{\mathrm{pg}}^{\mathrm{max}} = 0.4$, RL loss weight $w_{\mathrm{rl}} = 0.706$, and a moving-baseline EMA factor of 0.93 for variance reduction.

**Evaluation protocol.**   We evaluate through the SPEC-BENCH harness, which standardizes tokenization, decoding policy, batching, and logging across methods by testing them in the same environment and hardware. Competing methods are invoked via the official public implementations referenced by SPEC-BENCH (e.g., GitHub/Hugging Face integrations).

We do not retrain external baselines; when a method exposes a user knob (e.g., draft depth), we use the authors' recommended defaults as surfaced by the harness. Metrics follow the benchmark: mean accepted tokens (MAT) and end-to-end wall-time speedup relative to greedy autoregressive decoding of the baseline model.

**Absolute vs. relative speedups.** Despite using the official public implementations within a common harness, we observe absolute speedups that are lower than the numbers reported in the respective papers for all methods. Replicating on an NVIDIA A40 produced similar absolute speedups to the H100 in our setup, e.g. lower than reported in respective papers.

Importantly, the *relative* ordering is consistent with prior reports: EAGLE-family methods are fastest, followed by Hydra, then Medusa, etc. Our conclusions therefore focus on the apples-to-apples comparisons within the shared SPEC-BENCH environment and the relative efficiency of DVI under identical settings.

**Reproducibility** We will release code, configuration files (including split index, $k_{\mathrm{spec}}$, and decoding settings), and scripts to reproduce all tables and figures upon publication.

## B  AI ACKNOWLEDGMENT

We used AI-based assistants for stylistic polishing and editing language. All technical ideas, analyses, conclusions, etc. are our own.

