# OpenReview forum: "Draft, Verify, \& Improve: Toward Training-Aware Speculative Decoding"
_ICLR.cc/2026/Conference — Submitted to ICLR 2026_

### Official Review · Reviewer_r38y · 2025-10-23

**Soundness:** 2
**Presentation:** 2
**Contribution:** 2
**Rating:** 4
**Confidence:** 4

**Summary:**

This paper presents an interesting idea of integrating online learning into speculative decoding. However, the manuscript suffers from three major weaknesses that significantly undermine its contributions: limited performance gains, failure to address the core inference bottleneck, and outdated comparisons that ignore recent advancements like EAGLE-3.

**Strengths:**

​1. Novel Integration of Online Learning:​​ The core idea of closing the loop between speculative inference and online learning is innovative. Treating the verifier's commit decisions as real-time, self-supervised feedback for the drafter is a compelling approach to continual adaptation, potentially mitigating drafter brittleness under distribution drift without requiring separate offline datasets.

2. High Data and Training Efficiency:​​ A significant advantage of DVI is its minimal data requirement. The paper demonstrates that effective speedups can be achieved after exposure to only 2,000 prompts, which is substantially less than the millions of prompts required by methods like Medusa or EAGLE. This makes DVI a highly cost-effective and practical option for scenarios with limited training data or the need for rapid deployment.

**Weaknesses:**

1. Marginal Performance Improvements​
The claimed 2.16× average speedup appears modest when examined closely. As shown in Table 2, DVI's performance is actually ​inferior to EAGLE-2​ on several tasks (MT-Bench and Summarization), while the advantages in other tasks are minimal (e.g., only 0.07× faster in QA). Such marginal gains raise questions about the practical significance of the proposed method.

2. Misplaced Focus: Training Efficiency ≠ Inference Speed
The paper heavily emphasizes reduced training cost(using only 2,000 prompts) as a key advantage. However, this addresses a secondary concern while overlooking the primary challenge in LLM deployment: ​maximizing inference speed and minimizing latency.

3. Timeliness Issue: Missing Comparison with EAGLE-3
The most serious flaw is the omission of ​EAGLE-3​ (Li et al., 2024b), which represents the current state-of-the-art in speculative decoding.

**Questions:**

1 ​Include EAGLE-3 Comparisons: Essential experiments comparing DVI with EAGLE-3 under identical settings must be conducted.
2 Broaden Experimental Scope: Extend evaluations to larger models (e.g., 70B parameters) and different decoding strategies to demonstrate generality.

---

> ### Author Response · Authors · 2025-11-20
> **Addressing Reviewer r38y Concerns**
>
> Thank you reviewer r38y, we appreciate your feedback, have revised the paper accordingly.
>
> ## Weaknesses:
>
> > Marginal Performance Improvements
>
> We agree that maximizing inference speed and minimizing latency is the primary objective in SD. Our results in Table 2 demonstrate that DVI is not only competitive with the existing SoTA, but outperforms it on tasks like Translation, QA, RAG, and in the case of Vicuna-30B, even Summarization.
>
> While we agree that these advantages mean that EAGLE is still a competitive baseline, we believe that competitiveness itself denotes that DVI has practical significance, as our efficient training method is able to produce equally fast and robust speculative models relative to the SoTA using significantly less data.  Therefore, we believe DVI to be practically significant for many deployment settings as we give up almost nothing in inference speed gains while removing orders of magnitude of training cost and system complexity.
>
> We also wish to highlight the online nature of DVI - competing online methods like OSD are much slower than DVI. While relative to heavily trained methods like EAGLE, DVI's performance gains may not be as pronounced, DVI is significantly faster than other online methods. In other words, *DVI sets the new SoTA for online speculative decoding*, and as far as we can tell, is the *only* self-speculative method for online settings.
>
> $~$
>
> > Misplaced Focus
>
> We fully agree that minimizing inference latency is a primary concern in LLM deployment. Our emphasis on training efficiency is not intended to replace that objective, but to highlight a second axis that becomes equally important in realistic deployments.
>
> In many production settings, speculative drafters must be re-trained whenever the backbone, domain, or traffic distribution changes. In that regime, *speed is conditional on the cost of reaching and maintaining a calibrated drafter*.
>
> DVI is designed around this objective - it can match and surpass EAGLE level speed while ensuring that any necessary training or re-training is cheap and fast, enabling speedups that are robust to domain shift.
>
> Our online learning mechanism also means re-training does not require offline training or downtime - the training-aware design allows a deployed system to continually adapt its drafter to live traffic from live accept/reject feedback, keeping latency low as the distribution evolves.
>
> In the **revised version of the paper**, we now clarify this further in the introduction (Section 1, lines 60 to 66).
>
> $~$
>
> >Timeliness Issue
>
> Thank you for this suggestion. **In the revised version of the paper** we have added EAGLE-3 as a baseline where it is available. As the EAGLE-3 implementation is only released for Vicuna-13B (and not for Vicuna-7B or Vicuna-33B), we include a direct Vicuna-13B comparison between DVI and EAGLE-3 in Table 2. We see that DVI outperforms EAGLE-3 on 3 domains, while EAGLE-3 retains a higher average speedup relative to DVI.
>
> $~$
>
> ## Questions:
>
> > Include EAGLE-3 Comparisons
>
> As noted above, in the updated version of the paper, we include a comparison against EAGLE-3.
>
> $~$
>
> > Extend evaluations to larger models
>
> We agree larger experimental scope is necessary to demonstrate the generality of DVI. **In the revised version of the paper**, we expand our experiments to include 13B and 33B Vicuna models, as shown in Table 2. We continue to use Vicuna models for the reasons noted above - the Spec-Bench benchmark asks for a Vicuna backbone, and competing methods release Vicuna-based implementations.
>
> Across all 3 model sizes, we observe that DVI retains a competitive average speedup, while surpassing methods like EAGLE-2 and EAGLE-3 on Translation, QA, and RAG. In the case of Vicuna-33B, DVI also surpasses EAGLE on summarization.
>
> Due to compute limitations, we cannot train a 70B model and benchmark the other methods. However, we believe that our experiments across 3 model sizes is sufficient to demonstrate that DVI generalizes to multiple sizes.

---

### Official Review · Reviewer_GD6V · 2025-10-31

**Soundness:** 3
**Presentation:** 1
**Contribution:** 3
**Rating:** 6
**Confidence:** 3

**Summary:**

The paper proposes a training-aware self-speculative decoding framework that partitions a single LLM into shallow drafting layers and deep verification layers, then converts verify accept/reject signals into online supervision for the drafter.

A KL -> RL schedule warms up via online distillation to the frozen verifier and then adds reward-masked cross-entropy plus a light on-policy policy-gradient term, keeping speculation lossless under the verifier sampler.

On Spec-Bench with Vicuna-7B, DVI reports ~2.16X average wall-time speedup, competitive with EAGLE-2, while training on only 2k prompts and requiring no auxiliary drafter.

Overall, I think this is a good piece of work that provides insights in how to build novel speculative decoding frameworks and is worthy of acceptance.

**Strengths:**

1. The overall design is simple and deployment-friendly. The entire DIV consists of one backbone, a LoRA drafter head, and a frozen verifier.

2. The training-aware self-speculation turns commit decisions into online supervision, and is able to adapt the drafter to live traffic and mitigating distribution drift.

3. DVI is both data and computation efficient through empirical experiments. It achieves competitive speedups with a tiny online budget (e.g. 2k prompts; single-GPU setup), compared to orders-of-magnitude larger offline training for baselines.

**Weaknesses:**

Presentation needs to be improved. In fact that is the only factor that prevents this work from being published. For example, all equations are not numbered and reviewers are not able to refer to them. Some references are not in standard format, e.g. L74.

**Questions:**

1. How sensitive are results to split index k and proposal depth k_spec?

2. Can you report mean +- stdev over seeds for the main Spec-Bench tables? That should be the common practice for methods evaluated on Spec-Bench.

---

> ### Author Response · Authors · 2025-11-20
> **Addressing Reviewer GD6V Concerns**
>
> ## Weaknesses:
>
> > Presentation needs to be improved.
>
> We appreciate your feedback regarding the presentation of our submission, and we have made several revisions to address this. First, we now number all displayed equations and consistently reference them in the text (in particular, all equations in Sec. 3 are labeled and cited where used), so reviewers and readers can refer to them unambiguously. Second, we have corrected non-standard or inconsistent citations.
>
> If there are particular sections that still feel unclear or under-explained, we would be very grateful for pointers and will further refine the paper for the camera-ready version.
>
> $~$
>
> ## Questions:
>
> > How sensitive are results to split index k and proposal depth k_spec
>
> The results are somewhat sensitive to k and k_spec, as they act as design knobs for how the DVI model is built. Conceptually, they control the tradeoff between drafter strength (the deeper the split, the stronger), and per-step cost (the shallower, and shorter the speculation, the cheaper).
>
> A deep layer split index k will yield a stronger drafter, and higher MAT, but increases the cost of each draft forward.
>
> A larger k_spec raises the ceiling on accepted tokens per call, but also causes the draft path to run more times. If the drafter is expensive (like a deep layer split k), the added computation cost might dominate, negating any speedup. If the drafter is inaccurate, a large k_spec leads to rejected proposals and wasted computation, negating any potential benefit. However, if the drafter is highly accurate and cheap, this can yield a higher speedup.
>
> Thus, the results do depend on k and k_spec, but in a structured manner: shallow splits and small k_spec under-utilize speculation, and deep splits or large k_spec create extra computation and hurt speed.
>
> We follow the intuition and layer-selection heuristics used in prior self-speculation work such as Kangaroo. *In the revised version*, we clarify these choices more explicitly in Section 4.1 (Lines 299-303).
>
> $~$
>
> > Can you report mean +- stdev
>
> We appreciate the suggestion and agree that multi-seed reporting is good practice. However, the baseline implementations (EAGLE-1/2/3, Medusa, Hydra, etc.) are provided as fixed trained models rather than as quickly re-trainable recipes with multiple seeds. This is why existing Spec-Bench tables in the literature typically report single numbers for those methods.
>
> For DVI it is straightforward to re-run training with different seeds, but doing so for all competing methods and for all model sizes, and then evaluating each resulting model on Spec-Bench, is out of our compute budget.
>
> Retraining the competing methods locally can also introduce new issues like missing data / hyperparams / deprecated dependencies and so on, potentially causing the baselines to be trained incorrectly. That is why we choose to use official, public implementations of the competing baselines and methods.

---

> > ### Comment · Reviewer_GD6V · 2025-11-21
> >
> > Thanks for the response! I believe the current positive rating is a fair evaluation of this work. Good luck!
> >
> > Additionally, for my Q2, I think only inference-time efforts are needed, i.e., you don't need to train multiple models but only need to report speedup ratios with multiple runs. But again, that was totally optional.

---

### Official Review · Reviewer_STyx · 2025-11-01

**Soundness:** 3
**Presentation:** 3
**Contribution:** 3
**Rating:** 6
**Confidence:** 3

**Summary:**

This work introduce Draft, Verify, & Improve (DVI), a training-aware self-speculative framework that combines inference with continual online learning to tackle the training overhead of speculative decoding (SD) methods. DVI incorporates a frozen verifier and an online-learned drafter head that converts commit decisions of SD into self-supervision, making the SD model data-efficient without separate offline datasets or long pre-training. Experimental results demonstrate competitive speedups compared to other SOTA SD methods with minimal training overhead.

**Strengths:**

1. The idea of using a frozen verifier and an online-learned drafter head to save training overhead of SD models is interesting and promising. The presentation of the proposed method is clear and easy to follow.

2. The experimental results demonstrate the proposed method achieves competitive speedups compared to other SOTA SD methods with minimal training overhead.

**Weaknesses:**

1. The experiments is based on a small-scale, outdated LLMs Vicuna-7B. Further experiments on larger models (30B or 70B parameters) are expected to yield improved value of this work.

**Questions:**

Although the speedup metrics are competitive compared to other SOTA SD methods, there are still gaps between the proposed methods and SOTA methods in the mean accepted tokens metrics. How to understand this gap?

---

> ### Author Response · Authors · 2025-11-20
> **Addressing Reviewer STyx Concerns**
>
> Thank you review STyx for the feedback. We would like to clarify a few things below.
>
> ## Weaknesses:
>
> > experiments is based on a small-scale, outdated LLMs Vicuna-7B
>
> We used Vicuna as our primary backbone because Spec-Bench explicitly specifies Vicuna models as the standard base models for comparison, and the public implementations of Medusa, Hydra, EAGLE-1/2/3, PLD, and SpS are all built and tuned around Vicuna checkpoints. This choice ensures an apples-to-apples comparison across a broad set of SD baselines without re-training each method for a different model family.
>
> We cannot re-train each method due to compute limitations, especially given the amount of compute required by methods like EAGLE. We instead chose to use the implementations that were provided by the paper authors / official releases, often hosted on huggingface or on github.
>
> $~$
>
> > Further experiments on larger models (30B or 70B parameters)
>
> We agree further experiments were necessary to yield improved value for our work. **In the revised version** of the paper, we expand our experiments to include 13B and 33B Vicuna models, as shown in Table 2. We continue to use Vicuna models for the reasons noted above - the Spec-Bench benchmark asks for a Vicuna backbone, and competing methods release Vicuna-based implementations.
>
> Across all 3 model sizes, we observe that DVI retains a competitive average speedup, while surpassing methods like EAGLE-2 and EAGLE-3 on Translation, QA, and RAG. In the case of Vicuna-33B, DVI also surpasses EAGLE on summarization.
>
> Due to compute limitations, we cannot train a 70B model and benchmark the other methods. However, we believe that our experiments across 3 model sizes is sufficient to demonstrate that DVI generalizes to multiple sizes.
>
> $~$
>
>
> ## Questions:
>
> > gaps between the proposed methods and SOTA methods in the mean accepted tokens metrics
>
> Thank you for looking at our results in such great detail. To expand upon what we note in Section 4.1 (Lines 319-323), MAT is not a one-to-one predictor of wall-time speedup. You can increase MAT by (i) moving the split layer deeper or (ii) use more computationally intensive approaches like tree-decoding. These methods will increase the computation per step, but in turn create a more accurate prediction, yielding a higher MAT.
>
> Speedup, however, is dependent jointly on the acceptance rate, the draft/target cost ratio, and the proposal geometry, not just MAT alone. While competing methods yield a higher MAT, they also require more expensive drafting, limiting the added benefit of higher accuracy and negating potential speedup.
>
> DVI is intentionally designed in the opposite direction, where we use a very shallow layer and a small proposal depth to perform cheap, quick drafting. While we produce a lower MAT, each accepted token itself is significantly cheaper to compute, and therefore yields a higher speedup.
>
> In other words, DVI converts agreement into throughput more effectively due to the minimal overhead in drafting and verification. Some methods will naturally have a higher MAT than others, but that does not necessitate a higher speedup, as speedup is dependent on the efficiency of the decoding.

---

### Official Review · Reviewer_s2jc · 2025-11-05

**Soundness:** 2
**Presentation:** 3
**Contribution:** 1
**Rating:** 2
**Confidence:** 4

**Summary:**

The algorithm proposes Draft, Verify, & Improve (DVI) which is a self-speculative decoding method with online training. Specifically, it adopts first few layers as drafter and add lora head and use rest of the layers as verifier with verifier head. Then, it trains both lora heads in an online manner where supervision is given from the token acceptance reward. The result shows improved performance on Spec-Bench with smaller number of trainig data.

**Strengths:**

* The proposed algorithm introduces new types of online SD combining with self-speculative decoding.
* The motivation of the paper is clear.
* Proper ablations are conducted with sound presentations.

**Weaknesses:**

* **Novelty** : Prposed method combines self-speculative decoding with online training, but utilizing first few layers as drafter is already investigated as in [1]. Also, even DVI differs Online speculative decoding [2] in utilizing only accepted tokens, and combine it as a reward-signal in training, the effect of adding reward-suprevision is not independently investigated which limits the contributions of the paper.

* **Train time scaling** : While the proposed algorithm shows decent performance with only a samll amount of the train data, often one might need better drafter with more computes for training but no experiment is done.

* **Tree decdoing and stronger baselines** : The baseline should contain stronger baselines like EAGLE-3 [3] for fair comparison. Moreover, the result on tree-decoding of the drafter ([4], [5]) should be tested which generally shows improved speed-ups while the experiments are done only with single trajectory decoding.

* **Limited details** : Experiment details like warm-up steps or training hyper-parameters seems like being omitted.

**Questions:**

* Can authors show the performance of the SD along the number of trained tokens (i suspect the training might saturate earlier than other methods)?

* Can you test the trained model on tree-decoding scenario?

* Can authors evaluate the trained models on OOD dataset? I think RL-type component might hinder generalizability.


[1] (Liu et al.) Kangaroo: Lossless self-speculative decoding via double early exiting.

[2] (Liu et al.) Online Speculative Decoding.

[3] (Li et al.) EAGLE-3: Scaling up Inference Acceleration of Large Language Models via Training-Time Test

[4] (Cai et al.) MEDUSA: Simple LLM Inference Acceleration Framework with Multiple
Decoding Heads

[5] (Li et al.) EAGLE-2: Faster Inference of Language Models with Dynamic Draft Trees

---

> ### Author Response · Authors · 2025-11-20
> **Addressing Reviewer s2jc's Weaknesses**
>
> Thank you Reviewer s2jc for reviewing our paper. We have revised our paper accordingly, and would like to clarify a few more things below.
>
> ## Weaknesses:
>
> > Novelty
>
> We agree that the use of early layers as a drafter has already been explored by Kangaroo. We note that Kangaroo had explored this direction in the Related Works section (Section 2.2, lines 134-136), and we further acknowledged Kangaroo in Table 1.
>
> **In our revised version**, we further clarify the similarity in Section 3.2 (lines 207-208) by explicitly stating DVI adopts the same ‘early layers as drafter’ factorization, but we do not use Kangaroo’s dynamic draft length or early-exit mechanisms. Our contribution is orthogonal, and focused on the data-efficient training paradigm we create to enable self-speculation and online learning.
>
> Regarding the effects of reward supervision, we do independently investigate the effect of reward-based supervision in Section 4.3. There, we perform ablations to investigate the effects of using reward supervision (lines 419-426), distillation (lines 410 to 413), or cross-entropy (lines 428 to 431) independently. The results shown in Figure 2 and Table 3 demonstrate that KL alone is insufficient, and that reward-based terms are necessary to reach our reported speedups.
>
> In terms of online adaptation, as you note, we differ greatly from existing methods in the literature like OSD. OSD explicitly moves away from using a PG, only using distillation. Additionally, OSD relies on an offline warmup phase, whereas we do not. Our ablations (Fig 2, Table 3) demonstrate that distillation alone is insufficient, and our RL component is necessary to reach our reported speedup.
>
> Thus, DVI presents the first self-speculative method that treats the verifier’s accept/reject decisions themselves as training supervision for the drafter, enabling continual online adaptation.
>
> $~$
>
> > Train-Time Scaling
>
> Our goal is to study the low-training-budget regime for speculative decoding, where we want substantial speedups with minimal additional training. We demonstrate that with an extremely small budget, DVI still achieves speedups across all domains comparable to EAGLE-2, and outperforms EAGLE-3 on three of the six tested domains. This demonstrates that even with limited training, our drafter can outperform the SoTA without any heavy offline training.
>
> Our approach is not limited to a certain amount of training - DVI can be trained with more data or steps, but the novelty of our method lies in its data-efficiency. We therefore view a full scaling study over various larger training budgets as a complementary future work, rather than a limitation of our method. **In the revised version** of the paper, we make our intention regarding data-efficiency explicit (lines 60-66).
>
>
> $~$
>
>
> > Tree Decoding
>
> Thank you for this suggestion. **In the revised version** of the paper we have added EAGLE-3 as a baseline where it is available. As the EAGLE-3 implementation is only released for Vicuna-13B (and not for Vicuna-7B or Vicuna-33B), we include a direct Vicuna-13B comparison between DVI and EAGLE-3 in Table 2. We see that DVI outperforms EAGLE-3 on 3 domains, while EAGLE-3 retains a higher average speedup relative to DVI.
>
> In terms of tree decoding for DVI, we agree that tree decoding can further increase speedups when combined with a strong drafter. We already benchmark DVI against various tree-decoding methods (Medusa, Hydra, EAGLE-1/2/3), and observe that DVI outperforms many of them.
>
> The focus of our work is on building a training-aware, online self-speculative drafter, enabling cheap training and deployment. Extending DVI to also perform tree decoding would require substantial changes to the inference and training pipelines, which would obfuscate our central focus of compute-efficient online adaptation. Though we consider it to be out of scope for the paper, DVI’s training scheme could, in principle, be combined with a tree-based verifier in future work.
>
>
> $~$
>
>
> > Limited Details
>
> Thank you for highlighting this. **In the revised version** we update Appendix A to include more training details and hyperparameter configs.

---

> ### Author Response · Authors · 2025-11-20
> **Addressing Reviewer s2jc's Questions**
>
> ## Questions:
>
> > performance of the SD along the number of trained tokens
>
> We deliberately focus on the low-training-budget regime to show that DVI achieves speedups that are competitive with SoTA methods which are trained with orders of magnitude more data. Our intention is not to claim a particular ‘saturation point’ during training, but to demonstrate that when using DVI, substantial gains are obtainable with little data. As a byproduct, we can logically conclude that DVI saturates earlier than other methods due to our small training requirement, but that saturation point is heavily determined by hyperparameters.
>
> The shape of the learning curve is heavily influenced by scheduling (the KL warmup, the KL to RL ramp, the actual decay and anneal schedules) and hyperparameters (learning rate, PG weight, etc.). Therefore, the “how early” the model saturates is a design choice resulting from hyperparameters, and not an intrinsic property of DVI.
>
> Due to compute restraints, we do not do a full sweep of various training regimes and training lengths with DVI, and instead focus on the fixed-budget setting. We view a scaling study over more tokens or samples as a complementary future work.
>
>
> $~$
>
>
> > test the trained model on tree-decoding
>
> Adapting DVI itself to a full tree-decoding scenario (with multi-branch rollouts, tree-aware verification, and corresponding training signal) would require substantial changes to both the inference pipeline and the training objective, and would also significantly increase the training and inference compute—moving away from the simplicity and efficiency that are central to our contribution.
> We instead evaluate DVI against competing tree-based baselines such as Medusa, Hydra, and EAGLE-1/2/3, shown in Table 2. As we noted above, we find combining a DVI-style training strategy with a tree-style decoder to be an interesting orthogonal extension to our work, but out of scope for this paper.
>
>
> $~$
>
>
> > evaluate the trained models on OOD dataset
>
> We agree that checking if our RL approach generalizes to OOD evaluations is important. Our training data (ShareGPT) and evaluation benchmark (the six Spec-Bench domains: MT-Bench, Translation, Summarization, QA, Math, and RAG) differ in both task distribution and data source. Therefore, all of our reported Spec-Bench results are already out of training distribution, e.g. OOD.
>
> Across these domains, we see DVI consistently matches or tracks the SoTA methods in average speedup, even outperforming EAGLE-2/EAGLE-3 on several domains such as Translation, QA, and RAG. This empirical evidence demonstrates that the RL component does not harm generalization in practice. Moreover, our ablations in Section 4.3 demonstrate that the RL term is necessary to reach our reported speedup.
>
> **In the revised version**, we have explicitly noted that Spec-Bench acts as an OOD evaluation for ShareGPT-trained speculation methods (Section 4.1 lines 276-278).

---

### Author Response · Authors · 2025-12-02
**Rebuttal Summary for AC**

Thank you for overseeing the review process.

Across all four reviews, the only shared requests were (i) experiments on larger models and (ii) inclusion of EAGLE-3 as a baseline. In the revised paper, we now:
* Add results on larger Vicuna models (13B and 33B) alongside the original 7B
* Include EAGLE-3 wherever public implementations are available.

Below is a concise, reviewer-by-reviewer summary of the main concerns and how we addressed them.

---

$~$

# Reviewer s2jc
## Comment on Review Quality
We would like to note some concerns regarding the quality of reviewer s2jc’s review. Several of the stated weaknesses were already addressed in the original submission or appear to reflect misreadings of the paper: the RL component was already analyzed in our ablations; Spec-Bench was already an OOD benchmark; and our novelty was framed around our online training method, not the early-layer factorization. Likewise, data-efficiency is presented as a central goal, yet the review requests larger training budgets with little empirical justification, despite DVI already being on par with SoTA methods. Finally, the claim that our method “saturates earlier” than others is by design, and a strength we highlight as we train on 1000× less data than competing methods.

## Key Concerns
* Novelty relative to Kangaroo / RL investigation
* Lack of train-time scaling study
* Missing tree-decoding comparison
* Missing training details
* Training saturation
* Eval on OOD data

## How We Addressed Them
* Novelty:
**(i)** We expand upon our original discussion of Kangaroo and clarify that our contribution is not the “early layers as drafter” factorization (Section 2.2, lines 134-136, Section 3.2 lines 207-208), but the training-aware, online self-speculative paradigm.
**(ii)** We independently investigated the effect of reward-based supervision in our ablations (Section 4.3)
* Train-time scaling: We revise the paper to make our core goal of data-efficiency more explicit (lines 60-66). We believe a scaling study with larger budgets is out of scope since we demonstrate no need for a “better drafter with more computes for training”, as we are competitive with the SoTA.
* Tree decoding and stronger baselines:
**(i)** We add EAGLE-3 as a baseline wherever its implementation is available.
**(ii)** We note that implementing full tree decoding into DVI would require substantial changes to both inference and training, increasing compute and shifting the focus away from a cheap, training-aware, online self-speculative drafter.
* Limited details: We expand Appendix A to include training details and hyperparameters.
* Training saturation: We explain that DVI is deliberately designed for the low-budget regime—DVI becomes effective after only ~2,000 prompts. The “saturation point” is governed by schedules and hyperparameters, not intrinsic to DVI itself.
* OOD evaluation: We now explicitly state that Spec-Bench is already OOD relative to our ShareGPT training data (lines 276-278).

---

$~$

# Reviewer STyx
## Key Concern
- Experiments are based on a small, “outdated” Vicuna-7B backbone.

## How We Addressed It
- We explain that Vicuna is the standard model specified by Spec-Bench, and that all major baselines (Medusa, Hydra, EAGLE, etc.) are released and tuned for Vicuna. We choose Vicuna to enable a fair, apples-to-apples comparison without retraining every baseline.
- In the revised paper, we extend experiments to larger Vicuna models (13B and 33B) and show DVI remains competitive across these sizes under the same small training budget.

---

$~$

# Reviewer GD6V
## Key Concern
- Presentation quality (equations not numbered, citations inconsistent), noted as “the only factor that prevents this work from being published.”

## How We Addressed It

- We numbered all equations and corrected citations.

---

$~$

# Reviewer r38y

## Key Concerns

- Perceived marginal performance improvements relative to SoTA
- Focus on training efficiency rather than purely maximizing speed
- “The most serious flaw is the omission of ​EAGLE-3”

## How We Addressed Them

- Performance: We explain that DVI is not only competitive with SoTA methods but consistently outperforms them on Translation, QA, and RAG. It is also important to note the online nature of DVI - we are by far the fastest online SD method. We argue that our lower training cost and system complexity is practically significant despite competing methods having similar performance.
- “Misplaced focus”: We clarify that in realistic deployments, speculative drafters must be retrained whenever the backbone, domain, or traffic distribution shifts. In that setting, *effective inference speed is conditional on how cheaply one can (re)train a calibrated drafter.* DVI is explicitly designed for this regime - we revised the introduction to highlight this.
- EAGLE-3 / larger models: In the revised version, we add EAGLE-3 as a baseline where available and extend experiments to 13B and 33B Vicuna backbones.

---

### Meta-Review · Area_Chair_x9V6 · 2025-12-13

**Summary:**

Reviewers mainly have three concerns. 1) lack of novelty. 2) lack of comparison with Eagle 3; 3) only did experiments on a small scale. During the rebuttal, the author added a comparison with Eagle 3 on Vicuna-13B. However, the overall performance is still a bit worth than Eagle 3. Hence, after careful consideration, I tend to reject this paper.

**Reviewer Concerns:**

The reviewers mainly have the following concerns.
1) Lack of novelty: the idea of using the first few layers as the draft has been proposed by previous papers; even though the focus of this paper is slightly different, the overall novelty is limited.
2) Comparison with EAGLE 3: The results are added in the rebuttal process; however, the results are not encouraging enough.
3) Experiment scale: still there, but it is more like a limitation of the whole field rather than this paper's limitation

**Reviewer Scores:**

Based on the discussion of reviewers and authors, I do not see the evidence of reviewers to change their review.

---

### Decision · Program_Chairs · 2026-01-26

Reject